# An Adaptive Distributed Denial of Service Attack Prevention Technique in a Distributed Environment

**DOI:** 10.3390/s23146574

**Published:** 2023-07-21

**Authors:** Basheer Riskhan, Halawati Abd Jalil Safuan, Khalid Hussain, Asma Abbas Hassan Elnour, Abdelzahir Abdelmaboud, Fazlullah Khan, Mahwish Kundi

**Affiliations:** 1School of Computing and Informatics, Albukhary International University, Alor Setar 05200, Keddah, Malaysia; b.riskhan@aiu.edu.my (B.R.); halawati@aiu.edu.my (H.A.J.S.); 2Computer Science Department, Community College-Girls Section, King Khalid University, Abha 62529, Muhayel Aseer, Saudi Arabia; aselnour@kku.edu.sa; 3Department of Information Systems, King Khalid University, Abha 61913, Muhayel Aseer, Saudi Arabia; aelnour@kku.edu.sa; 4Computer Science Department, Abdul Wali Khan University, Mardan 23200, Pakistan; fazlullah.mcs@gmail.com (F.K.); mkundi@awkum.edu.pk (M.K.)

**Keywords:** DDoS attack, SYN attack, attack mitigation, security

## Abstract

Cyberattacks in the modern world are sophisticated and can be undetected in a dispersed setting. In a distributed setting, DoS and DDoS attacks cause resource unavailability. This has motivated the scientific community to suggest effective approaches in distributed contexts as a means of mitigating such attacks. Syn Flood is the most common sort of DDoS assault, up from 76% to 81% in Q2, according to Kaspersky’s Q3 report. Direct and indirect approaches are also available for launching DDoS attacks. While in a DDoS attack, controlled traffic is transmitted indirectly through zombies to reflectors to compromise the target host, in a direct attack, controlled traffic is sent directly to zombies in order to assault the victim host. Reflectors are uncompromised systems that only send replies in response to a request. To mitigate such assaults, traffic shaping and pushback methods are utilised. The SYN Flood Attack Detection and Mitigation Technique (SFaDMT) is an adaptive heuristic-based method we employ to identify DDoS SYN flood assaults. This study suggested an effective strategy to identify and resist the SYN assault. A decision support mechanism served as the foundation for the suggested (SFaDMT) approach. The suggested model was simulated, analysed, and compared to the most recent method using the OMNET simulator. The outcome demonstrates how the suggested fix improved detection.

## 1. Introduction and Background

### 1.1. Introduction

The introduction of the attack through machines is known as “Supplementary Victims”, while the in-attack routing protocols are “Main Victims”. In this case, tracking the attacker is becoming difficult taking into account legitimate customers.

Network security has become a complex challenge for companies with a data centre or network configuration. Various hardware and software resources are used to unencrypt passwords from several attacks. The push-back procedure is widely used to support distributed denial-of-service attacks. DDoS activities are viewed as a traffic control issue using a router, even though the deceptive controller causes disruption and therefore does not manage congestion after the conventional edge. The newest generation of the router is sensitive enough to detect and lower suspicious packets. Onshore routers have been informed of the decrease in suspicious packets and advised to use router services for legitimate content instead. A further unique and reliable countermeasure for handling DDoS attacks is the implementation of user riddles [1,2]. Throughout this strategy, the victim system challenges a system that sends traffic to recognise the attacker. If the client solves the mystery, the traffic is viewed as valid and thus expected to move to the client; unless most of these people solve the riddle, the difficulty of the riddle is enhanced. Such a strategy ensures the continuous flow of network traffic across the intermediary routers before it reaches its destination (Figure 1) [3].

A single-layer rational system is a circular base feature system used to diagnose irregularities and classify regular traffic. These proposals add ingenuity to Denial-of-Service prevention, although information-based positioning is often used. The simple advancement of signature-type attack identification is commonly used when inbound traffic is likened to accessible, recognised strikes called white-list patterns (information). It efficiently detects potential threats throughout the signature data set [4].

One strategy for detecting DDoS is attitude-based identification, which can distinguish DDoS-attack traffic from sanctioned traffic irrespective of different ways of attacking content and techniques. Currently, DDoS strikes are conducted using tools, worms, and botnets to victimise entirely different packet transmission rates and packet aspects of the defence strategy [5,6].

As a result, these different types of attacks contribute to defence systems requiring other encryption methods on-site. DDoS attacks aim to render traffic unavailable, including Flash crowd cases. The findings of experimentations with several databases and tests suggest that the predicted techniques can separate DDoS threats from lawful traffic [7]. Denial of Service has emerged as a major threat to several companies nationwide. DoS attacks are resolved through the series number encryption technique, and the hop sequence filtration approach efficiently filters attack packets, providing the database with appropriate security [8].

To secure two-layer protection strategy resources, it is suggested that the MAC generator be isolated legally from the encrypted one, through which the client services are distributed to legitimate lanes and lawful customers efficiently [9,10]. Traditional methods for detecting a distributed denial of service (DDoS) attack have also been unsuccessful. Throughout this journal, artificial neural systems and clustering algorithms have been suggested for a new compact tracking strategy. At the same time, the ANN Multi-layer Perceptron has been used to enhance conviction rate and precision [6,11]. The outcome of the whole analysis is influential compared to earlier studies and a fantastic way forward towards future research [12]. The aim is to identify and prevent specific DDoS attack trends and strategies from occurring in a decentralised setting. It is a remedy for the identification and mitigation process, wherein the SFaDMT methodology works in a single-node activity. SFaDMT can be used efficiently to identify sequence recognition and signatures that already occur throughout the SFaDMT system [13,14,15].

Once a DDoS intrusion is performed on a system, the application of resources to potential users cannot be successfully achieved. In order to address this problem, it is suggested that DDoS identification, as well as prevention techniques, be referred to as SFaDMT. The whole strategy describes the SYN Flood attack on the system and minimises it to execute streamlined behaviour for the system [10,16,17].

One crucial method for stopping cyberattacks is intrusion detection, which may be divided into three categories: hybrid detection, misuse detection, and anomaly detection. For example, anomaly detection uses network data and connection traffic to find threats and typical access behaviours. However, traditional behaviour identification-based anomaly detection is unable to meet the demands due to the large-scale, dispersed, and non-standard physical components present in ICPS and IIoT.

The heavy computational burden of cloud data centres and the monitoring of anomalous access to physical units with set communication cycles are two technological issues that require attention from a federated learning technique that decentralises the detection work to the edge, considering the former [18]. Knowing how a cyberattack is designed is the most crucial factor in a CPS’s security. Knowing the structure of such a cyberattack is a crucial component of a successful mitigation plan for the security of CPS.

A variety of cyber-attacks were developed against CPS components to explore this, and the impact on cyber, physical, and collaborative control components was assessed. Stuxnet [16] and the Aurora assault [19] raised awareness of and sparked widespread worry about cyberattacks that may harm physical infrastructure. As previously said, since most current security measures were created for cyber-only systems, they cannot be easily applied to CPS in a collaborative network. New strategies are required to stop CPS failure. The interface is a crucial node where cyber components enable a wide range of assaults due to the differences in the physical and cyber layers’ features inside CPS. The PC, in comparison, is rigid and straightforward, with very few attack alternatives [20].

### 1.2. Literature Review

Pushback is a strategy used to defend against DDoS attacks. DDoS attacks are mostly successful because traffic can be carried out with malware hosting in the decentralised system, and end-to-end traffic management cannot be conducted and can be managed by a function in the new router. The packets related to the intrusion must be identified but most likely contributed to the strike [21,22,23].

To complete just the lawful traffic’s progress upward, routers will inform of the cancellation of the deceptive traffic. In certain cases, the user question has been used as a common strategy for the past few years to help alleviate the DDoS attack [21]. The target system assigns a riddle to the end user to define and discriminate between legitimate and deceptive traffic. If the user effectively solves the riddle, it is presumed that the user is a legal end-user, and permission to access the database will also be given. Unless the highest possible number of clients can overcome the riddle, the system may increase the difficulty of the riddles. When it hits the end state, it is a crossroads for malicious information [24].

The strategy for detecting DDoS using actions-based identification can distinguish between distributed denial-of-service (DDoS) traffic and legal traffic, irrespective of the various types of intrusion transmissions, including techniques [14,25,26]. Today, DDoS attacks use software, worms, and botnets to victimise entirely different transmission rates and packet types to defeat defensive systems. Accordingly, these different types of attacks contribute to protection systems offering other detection systems for ground attacks. DDoS attacks go through traffic like Flash population cases [27].

DDoS attacks include options for reproducible variations that unite the area separately from the normal crowd flow of traffic. In this journal, similar detection approaches have been used to endorse Pearson’s statistics. Techniques can derive reproducible options from packet deliveries within the DDoS traffic, not from quick crowd congestion. Comprehensive models have been conducted to enhance detection systems [22].

The results of the experimentation have been shown regarding many databases, and our findings support the predicted techniques by which DDoS attempts could be distinguished from legal traffic [23,28]. Denial-of-Service attempts are a significant downside for the tech community, given that the research group has also developed a comprehensive scope of security strategies.

Throughout this journal, we aim to implement information technology’s rapidly hopping, easily remotely operated, and efficient channel-layer architecture against DDoS attacks. Our solution provides a clear method for potential buyers to protect the functionality and target database of the correspondence activities. We tend to describe the Dynamic Database Server Address Alteration technique, but each component implements the approach [12].

DDoS flood-based packet strikes are a very common technique and are successful against the accessibility of facilities and apps on the system. They are quite hard to detect and avoid due to the decentralised framework. The new technology addressed throughout this journal is Stop-It. Throughout this methodology, combative processes premised on filters are prepared to prevent attacks from happening. Big DDoS floods are centred on assaults. Nevertheless, this could be unsuccessful unless the concentration connection is communicated to the survivor. The journal shows a clear variation of the Vary system within the Stop-It methodology. Directly and indirectly, attacks can be controlled to minimise DDoS attacks [13].

Throughout this journal, the author points out how GET Flood’s interaction mechanism is incorporated into distributed denial-of-service attacks for rapid attack identification in a decentralised setting. By contrast, interval simulations are performed to align efficiency with the trend identification of attack alternatives and Snort identification of approved communications protocol stream trends, including log data from a network server. Experimental data indicate that the proposed strategy is safer than the identification of Snort because the previous period was smaller for that traffic. Furthermore, the whole strategy will ensure the ability of the target computer to be associated with the preventative and dependable identification of endorsed information and communication procedures [14].

DDoS strikes send large amounts of network traffic to the target system through the victimhood of various systems. Flow-based object detection strategies have performed significantly better than fingerprint-based attack identification techniques in these tests. Flow-focused DDoS attack identification methods were separated into two classes, i.e., packet-header-predicated and numerical-implementation-based. In that job, the goal is to examine each computational principle to investigate the DDoS attack mechanism and to maintain false pros and cons focused on problematic control bench victimhood advanced systems.

The journal has also been evaluated and tested in terms of precision, including the ability to perceive, and its development is recommended to produce even better outcomes than the two algorithms initially proposed as different strategies:

Signers based;

Anomaly related;

DNS related;

Mining cantered.

A comparison, including an examination of the benefits and drawbacks of the approaches alluded to here, can be made. Throughout the current situation, however, no one discussed the issue of why it is hard to detect current botnets and how we might utilise fluxing strategies to detect them. Throughout this research, two more sophisticated botnet-level strategies are mentioned: Fast-Flux-Single-Flux and Double Flux-Domain-Flux-Torpig (FFSN), which passive and active strategies could identify.

First, the author suggested a DNS-based RDNS monitoring strategy for detecting unauthorised flux system networks throughout this journal. Second, the flux agent surveillance system consists of four elements. To obtain information and add new IPs to the IP track repository, a new technique was created throughout the title of the Dig-Tool; the key element was the tracking agent, which delivers the HTTP server to the IP track repository, and that same reaction is reported. The final aspect is an IP lifetime records server for recording the system’s condition, i.e., “1” for the system being available, whereas “0” is for the service not being accessible.

## 2. Proposed Solution and Methodology (SFaDMT)

The Internet Protocol is a cloud-interface communications server that is a delivery service for packets. UDP is not that efficient, which implies that perhaps the distribution of packages is not guaranteed. However, network-less implies that certain packets could keep their own records and be independent of transmissions. The Transmitting Control Interface can be improved with a one-sided network layer and the Protocol on another. Accurate contact between applications and different networks is assured. TCP allows the efficient transmission of data streams in sequence with no duplication or malfunction.

### 2.1. Establishment of TCP Connection—(Three-Sided Shake of TCP)

TCP connectivity is formed with a three-sided handshake. First, the user delivers link queries by submitting the SYN message for the host; the host acknowledges this by sending the SYN to acknowledge packets back to the recipient node, which also distributes the contact space throughout the queue. Eventually, the user accepts the ACK packets and the communication phase is finished.

### 2.2. The Technique for DoS/DDoS Outbreak (TCP-SYN Flooding Intrusion)

A TCP (Transmission-Control Protocol)-synchronised flood intrusion is a harmful DDoS/DoS attack initiated by an assailant via several connections. In these connections, SYN-ACK and SYN packages are swapped frequently, resulting in a shortage of ACK messages that are not sent to the server. As a result, the system leaves demilitarised space dedicated to all unfinished links, so there is no space available for anti-malicious link queries that prevent end-users from using the survivor scheme or channel. The SYN Flood strike is based on a three-way sequence of technical handshakes that starts transmitting control procedure associations. Throughout this grouping, the third package indicates the initiator’s capacity to retrieve packets at the IP address and its initial message, which utilises the origin or restores the retrieval capability. Figure 2 describes the start of a standard TCP link transmitted at the packaging chain. Link details are preserved by a collection of mechanisms throughout the operating system and the transmission mechanism code framework in the Transmission Controls Box (TCB).

The TCB storage capacity depends on the TCP settings, the functionality offered by the configuration, and the communication allowed. The Transfer Control Box had 282 bytes and 1300 bytes throughout the new OS. The transmitting control procedure’s collected status synchronisation suggests that the contacts are just half-open. At the same time, the validity of the demands remains a question to be answered. The main point would be that the transmission-controlled box is distributed based on the synchronising package obtained before the link is created or the initiator’s accessibility is verified.

It is a good indication of the denial of service that the receiving synchronisation is now the distribution that triggers the distribution of the transmission controls procedure. This can also deplete the capacity of the server processor. The aim of the Transfer Control Policy flood synchronisation intervention is to reduce the delay via the synchronisation sections, which occupies the full bottleneck. Link-encrypted communication domains are used by synchronisation assailants and do not trigger any reaction to line the Transmitting Control Box when using the obtained synchronisation classification. The transmitting Control Procedure aims to be credible, so servers keep their Transfer Control Blocks in synchronisation for a longer period until the two-thirds connection has been released. Meanwhile, the network is denied entry demands for legally permitted transmission control procedure connections.

Figure 3 specifies a set-up of synchronization inundating attacks and provides a general clue for such interventions.

Figure 4 shows some dissimilarities which have been observed.

## 3. Proposed Technique (SFaDMT)

SFaDMT will filter the SYN packets from incoming traffic throughout the projected strategy, and rules are implemented in the SFaDMT to detect the DDoS attack signatures.

### 3.1. Flowchart

Once traffic comes up, the signatures are compared to the current repository. If paired, the traffic has been considered malignant and would be obstructed from obtaining access to the system. If the traffic sequence does not suit the signatures still in existence in the computer system, it will be entered in the SFaDMT, where a contrast is made between the current signatures and the traffic that comes in the SFaDMT. If the signatures have a correspondence here between two traffics of more than 71%, the inspection of the system will be allowed and will be deemed harmful. If the correspondence is lower than 69%, this will be viewed as a legal flow of traffic and will be able to obtain access the network effectively. An adapted solution for the identification of DDoS intrusions in a centralised setting (Figure 5): (a)Application of a multi-deposit preceptor deposit in a centralized environment.(b)Pattern and signature-centred strategies are used to identify DDoS intrusion.

### 3.2. The Model of SFaDMT

An inspection was carried out on the traffic coming to extract SYN messages. Subsequently, as in the suggested solution, the signatures could be compared with the traffic to decide whether the traffic is authentic or deceptive. When unauthorised communication has been identified, the SFaDMT will be informed and traffic will be stopped from entering the system. It will alert, review the repository, and create logs at any time, but if the exact type of attack happens on the system, it will be identified after the contrast. If there are fingerprints, if it does not meet the SFaDMT fingerprints or if it has less than 71% correspondence, it will be regarded as legitimate traffic. This will upgrade the SFaDMT and allow you to obtain a connection effectively. In the context of traffic, the behaviour is unclear. A layer packet examination identifies traffic and determines whether it is deceptive or legal. The channel would be reached whether the transmission is legal or not, while the deceptive traffic would be restricted (Figure 6).

### 3.3. Proposed Solution and Methodology

The recommended strategy introduces a method when the transmission collects at the channel portal, as seen in the illustration described. Afterwards, the recommended strategy protects it from disruptive activity. Malignant traffic may be distinguished through signature-centred detection. A multi-layer strategy is used throughout this method to suit the identities currently existing on the server. If the signature indicates some similarities, a detailed description will be carried out in the SFaDMT system, and even if the outcome suits 71% of the accessible signatures, the fraudulent traffic will be mitigated and isolated from the system (Figure 7).

### 3.4. Guidelines for the Categorization of SYN Flood Malevolent Instructions/Apps

In the proposed technique, a hybrid solution is presented, as shown in the above image. We have deployed a solution that will protect the packet from malicious traffic as it arrives at the network gateway. The malicious traffic can be isolated using signature-based detection. In this solution, a multi-layered approach was used in which the signatures were matched with the already-present signatures in the database.

A detailed analysis will be performed in the SFaDMT module if the signature is similar. If the result matches 70% of the signatures available, it will mitigate that malicious traffic and isolate it from the network. The filtration rules of SYN flood packet are described below, whereas Algorithm 1 is presented the procedure to detect half-open TCP connection, Algorithm 2 presented how to detect malicious traffic based on signatures in already archived attacks signatures and finally Algorithm 3 the mitigation mechanism is described the working of SFaDMT.
**Algorithm 1.** Check for Half Open TCP Connection.whileread present connection;if (connection attempt is not successful ||TCP connection is not synchronized at both ends ||TCP connection is aborted || connection cannot be closed)     thenthe TCP connection is malicious;      elsethe TCP connection is legitimate;   endend

**Algorithm 2.** The following algorithm implements the above rules and detects malicious traffic based on signatures from the already-saved attack signature database.Input: **Packet Pkt**Output: **Generates logs when DDoS Attacks are performed**
**while (true)**
**if** (a packet is not equal to null) **do****if** (TCP packet arrives) **do**TCPCount++;**if** (TCP Packets % threshold ≥ 70) **do**Alarm “TCP Flow Attack has been detected!”;**else if** (IP Packets % threshold ≥ 70 && source_ip == destination_ip) **do**Alarm “IP Address Pattern has been detected!”;
**else**
Show “Error”       **end if;**
     **end if;**

**end if;**


**Algorithm 3.** The following algorithm is the mitigation of attack traffic performed using SFaDMT.Input: **Data Traffic Dt**Output: **Analyze Dt**
**while (true)**
**if** (Dt arrives) **do**Dt++;**if** (Packet Header == Legitimate Dt) **do**Display “Legitimate Traffic and can access the network”;**else if** (Packet Header == Malicious Dt) **do**Display “Malicious Traffic has been detected and mitigated”;**else if** (Packet Header == Unknown Dt) **do**    Display “Deep Analysis needs to be performed on this packet”;**else if** (Packet Analysis == Dt Attack Pattern Detected) **do**Display “Restrict traffic from the network”;
**else**
Display “Legitimate traffic has been detected and mitigated”;

### 3.5. Rules for the Filtration of SYN Flood Malicious Packets

Rule 1 → {SRC_IP ≠ DST_IP}

Rule 1 allows only packets with different source IP addresses and destination IP addresses. However, it will consider attack traffic if it finds the same source and destination IP address.

Rule 2 → {SRC_PORT ≠ DST_PORT}

Rule 2 allows only TCP packets under the condition that the source port must not be equal to the destination port. This is because it will be treated as attack traffic if it finds the same source and destination port.

Rule 3 → {tcp.FLAG = SYN}

Rule 3 investigates the TCP packets that are the SYN packets for further analysis by SFaDMT.

## 4. Result and Analysis

A simulator was built on a high-end system accessible in a computer lab to predict congestion for analysis and study the suggested structure (SFaDMT). The topology was built in a modelling tool called OMNET++, with nodes and clusters ranging from 20 to 300 mobile users. A distinction was made between the suggested adaptive approach and the pushback. During the execution of both strategies, a traffic dive was created repeatedly by different situations. The empirical study of simulation-generated outcomes could be seen in the net parts of the whole section.

The OMNET++ simulator was used to produce tests. A digital topology was developed, where nodes varying from 20 to 300 were only used to produce traffic flashes in the system, and the suggested dynamic methodology was used to identify SYN Flood attacks. The outcomes attained can be seen in the graphs of the output graph, which show the identification contrast between pushback and the suggested SFaDMT method. This indicates that the SFaDMT technique demonstrates that the results are faster and more efficient than those using the pushback methodology. The produced digital topology can be seen in the preceding images, below.

### 4.1. Case I

During the first test, as seen in Figure 8, a topology of eight nodes or modules was generated to implement the suggested SFaDMT methodology. Then, a traffic burst was produced and the recommended methodology was used to identify a DDoS SYN Flood strike. The suggested strategy recognised the SYN Flood invasion at one megabyte, while the pushback strategy recognised the SYN Flood strike at 9–10 megabytes.

### 4.2. Case II

In Case II, as seen in Figure 9, a topology of 40 nodes and modules was developed for the simulated world of the SFaDMT methodology. With even more entities throughout the topology, a greater blast of traffic was produced relative to Case I. The suggested methodology identified a DDoS intrusion at 3–4 MB, while the pushback strategy detected DDoS attacks at 14–15 MB.

### 4.3. Case III

As seen in Figure 10, a topology of 120 modules or nodes was generated for the simulations of the suggested SFaDMT methodology during most of the fifth cycle. A traffic burst was produced, and the applied methodology was used to diagnose a DDoS SYN Flood strike. Because the nodes increased for each situation throughout the topology, a greater congestion burst was created compared to the prior case. The suggested methodology identified a DDoS intrusion at 14–15 MB, while the pushback strategy detected a DDoS attack at 40–43 MB.

### 4.4. Case IV

A topology of 300 nodes or modules was developed, as seen in Figure 11, for the simulated model of the suggested SFaDMT methodology. A traffic burst was produced, and the proposed procedure was used to identify a DDoS SYN Flood invasion. The level of traffic burst improved, given the number of points and nodes. The recommended strategy recognised a DDoS attack at 11–12 MB, while the pushback strategy recognised a DDoS invasion at 60–63 MB.

## 5. Evaluation Parameters

To evaluate the proposed SFaDMT Model simulation, a test bed was formulated according to the following parameter details described in Table 1.

### Analysis of the Adaptive Technique with the Previous Technique

The legitimate traffic analysis graph generated traffic to analyse the good- and bad-will packets. It is shown in the graph during the first run that the push-back technique detected the traffic, including malicious packets at 1.2 MB. In contrast, the adaptive SFaDMT technique detected at 1.0 MB. Therefore, whereas legitimate traffic was 0.8 Mb, the traffic detection ratio of the adaptive SFaDMT technique was better in the performance analysis than the pushback technique.

Figure 12 shows that during the second run, while generating the traffic pushback detection, the malicious traffic in the graph was at 1.8 MB while the SFaDMT identified it at 1.3 MB. Similarly, on the third run, the pushback technique detected at 1.75 MB, and SFaDMT identified at 1.4 MB. In the fourth run, pushback detected at 1.9 MB, whereas the adaptive proposed technique detected at 1.5 MB. In the fifth run, the pushback detected at 1.9 Mb, and the adaptive technique detected at 1.7 Mb, clearly showing the performance increase ratio between the previous and adaptive techniques. Finally, at the sixth run, both techniques almost detected at 1.9 MB. Still, the SFaDMT technique detected the traffic, including malicious packets, earlier than the previous technique used in the graph for comparison. For contrast, the details of those runs are shown in Table 2.

The seventh sun pushback technique detected at 1.75 MB, and the proposed technique detected at 1.65 MB. The eight-run pushback technique detected the good- and bad-will packets at 2.1 MB, while the SFaDMT identified them at 1.9 MB. In the ninth run, SFaDMT detected the malicious traffic that flows with the legitimate traffic at 1.7 MB, whereas the pushback technique detected it at 1.9 MB. Hence, it was concluded that there were fewer false positives in the proposed SFaDMT technique than in the previous technique, known as the pushback technique.

The detection comparison between the pushback and SFaDMT techniques shown in Figure 13 shows that during the first run, the pushback technique detected the DDoS attack at 10 MB. In contrast, the SFaDMT detected the attack when only 1 MB of traffic was generated.

On both the second and third runs, the SFaDMT technique similarly detected the traffic at 1 Mb, whereas the pushback technique rose to 14–15 Mb. The complete details with runs are shown in Table 3. As the burst size of traffic increased in each run, the SFaDMT detection technique detected the attack at 3 MB, while the pushback rose and reached 20 MB. In the fifth run, the SFaDMT detected at 14 MB and the pushback detected at 43 MB.

In the sixth and last run, the SFaDMT detected at 11 MB, whereas the pushback technique detected at 63 MB, which shows that the adaptive technique can detect an attack at an earlier stage and increase the performance of the detection technique.

## 6. Conclusions

Among the most prevalent DOS (Denial of Service) attacks is the Syn Flood strike. It can affect the business side and services for legitimate customers, and it is impossible to eradicate such an invasion. However, the proposed strategy can considerably decrease the risk and harm caused by such attacks by taking the initiatives outlined in the journal. In a paper, the authors suggest an adaptive DDoS tracking mechanism that activates and minimises the attacks of Syn Flood. The envisaged SFaDMT methodology streams the TCP packets and the tracking contrast, which is decided based on the policies based on the registrations characterised throughout the SFaDMT identification system. The attack is observed if the network traffic ratio is greater than 70% relative to the regulations, and the signatures are processed. The traffic is deemed suspicious relative to the host network and is prohibited from accessing system resources. When the contrast is less than 75% of the traffic from a different host, it will be treated as encrypted traffic. If the traffic sequence is unclear, a layer packet examination can be conducted on the SYN packet interface to determine if it includes suspicious content. A prototype was built in OMNET for the application of the SFaDMT methodology with 20–300 mobile users, as well as a contrast with the pushing-back strategy. Traffic loads of varying sizes were created for identification, including network traffic assessment using the suggested adaptation and pushback techniques. Tests have shown that the new methodology has a 26 percent better identification rate than the current push-back strategy.

## Figures and Tables

**Figure 1 sensors-23-06574-f001:**
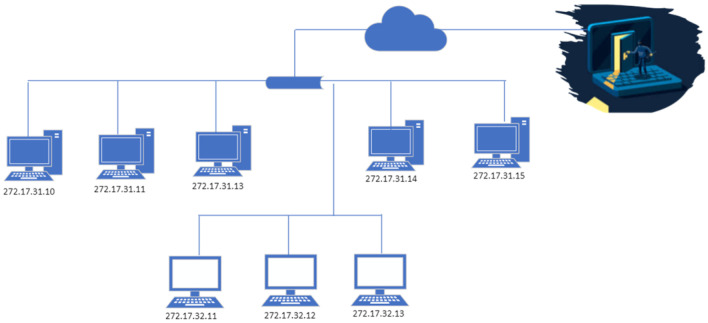
DDoS attack scenario.

**Figure 2 sensors-23-06574-f002:**
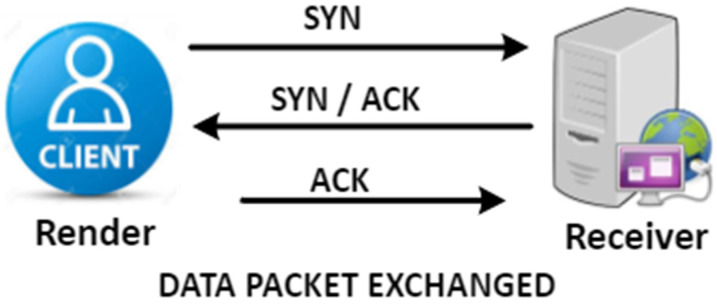
Three-way handshake of TCP connection.

**Figure 3 sensors-23-06574-f003:**
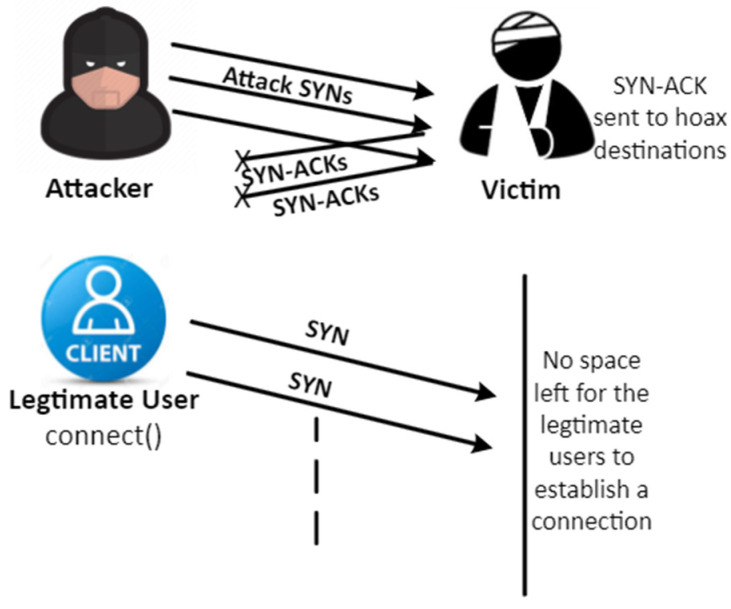
Denial of request to legitimate user.

**Figure 4 sensors-23-06574-f004:**
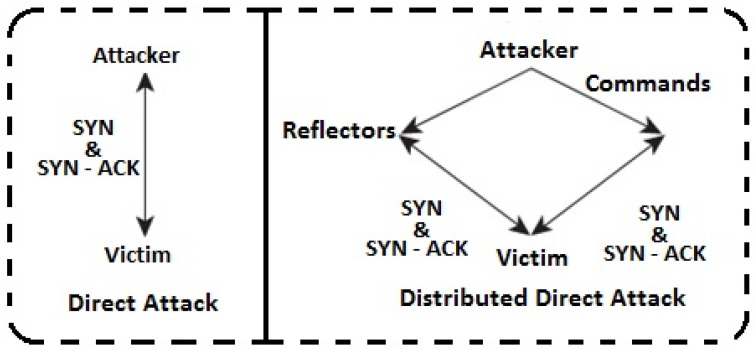
Types of basic DDoS attack.

**Figure 5 sensors-23-06574-f005:**
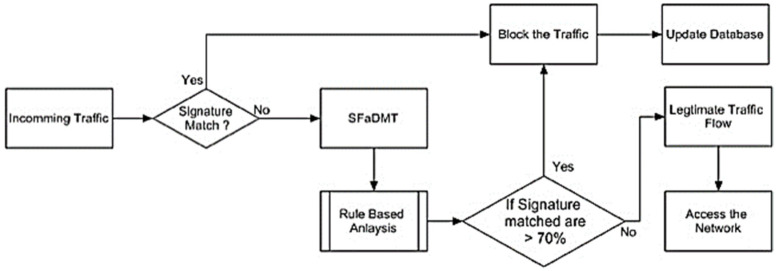
Flow chart of SFaDMT technique.

**Figure 6 sensors-23-06574-f006:**
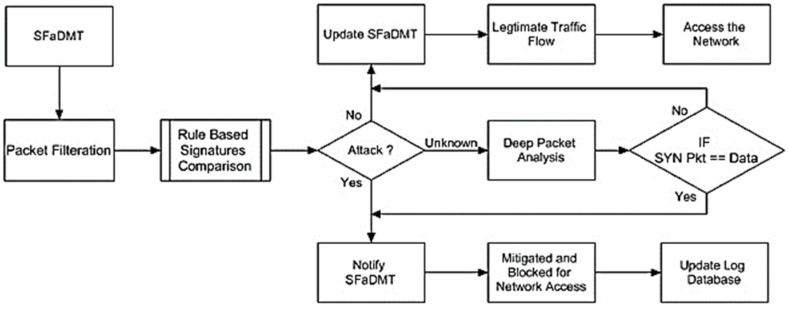
Framework of SFaDMT technique.

**Figure 7 sensors-23-06574-f007:**
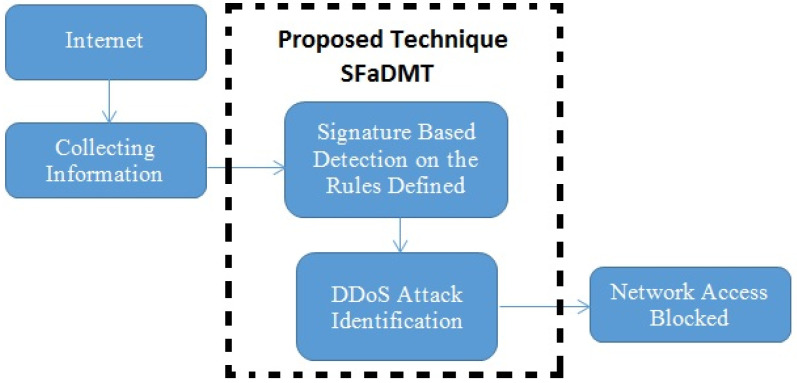
Proposed SFaDMT solution for detection of DDoS attacks.

**Figure 8 sensors-23-06574-f008:**
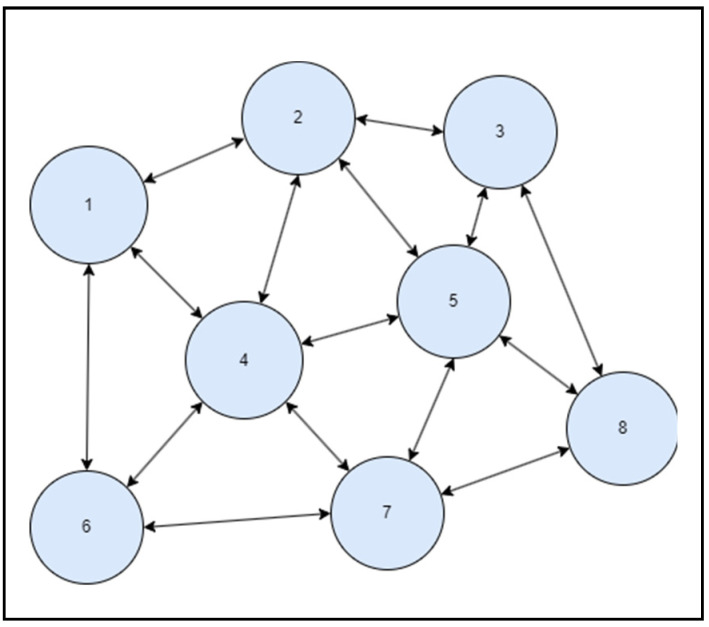
Topology of SFaDMT during the first run.

**Figure 9 sensors-23-06574-f009:**
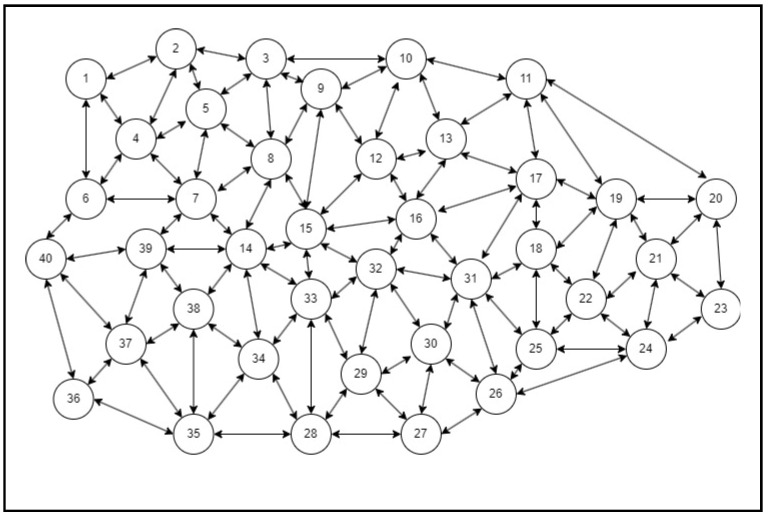
Topology of SFaDMT during the third run.

**Figure 10 sensors-23-06574-f010:**
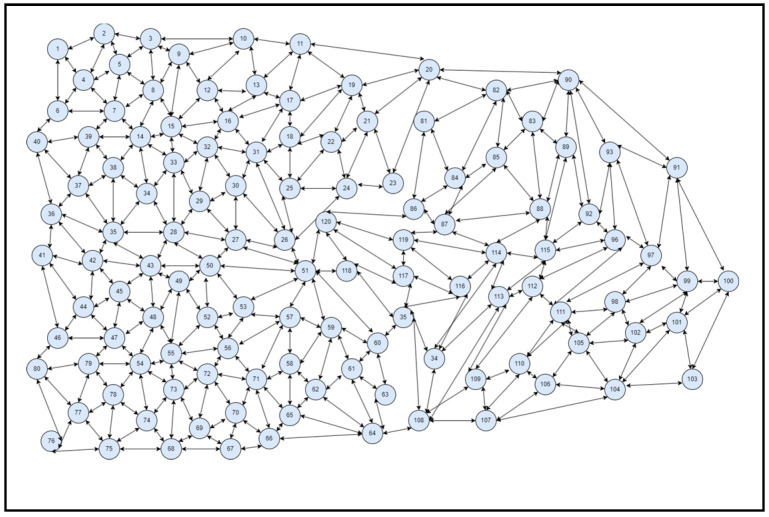
Topology of SFaDMT during the fifth run.

**Figure 11 sensors-23-06574-f011:**
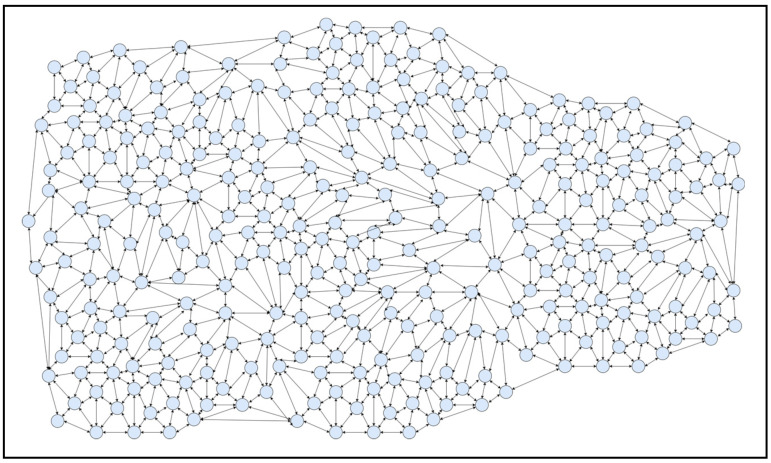
Topology of SFaDMT during the sixth run, with 300 nodes.

**Figure 12 sensors-23-06574-f012:**
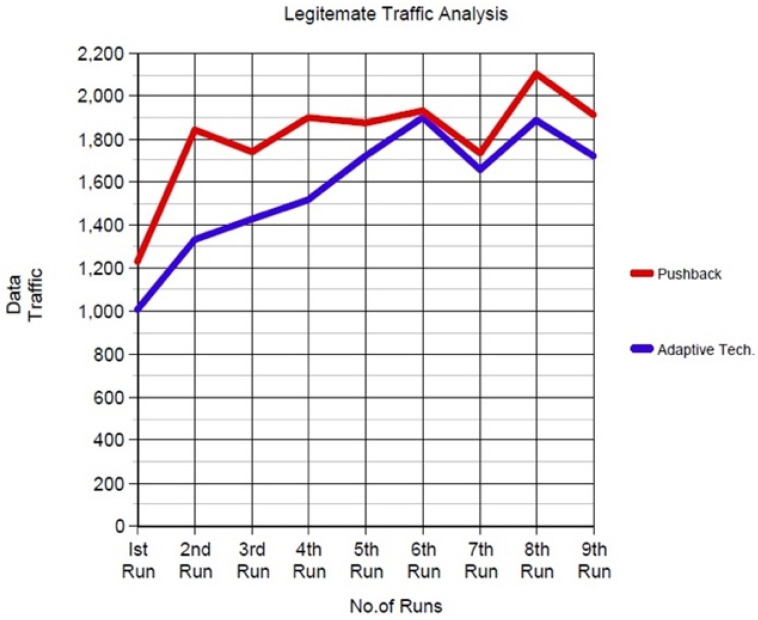
Graph comparisons of SFaDMT and pushback technique.

**Figure 13 sensors-23-06574-f013:**
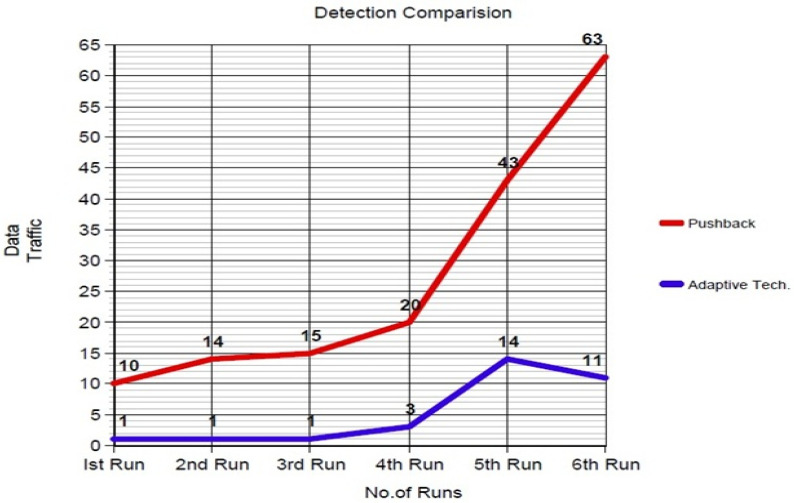
Detection comparisons of pushback and SFaDMT technique.

**Table 1 sensors-23-06574-t001:** Evaluation Parameters for OMNET Simulation.

Evaluation Parameters	
Nodes	10–200
N/W Type	Static
Traffic Burst	1.0–2.2 Gb
Malicious Node	Unknown
No. of Run	1–9

**Table 2 sensors-23-06574-t002:** Traffic comparison analysis of both techniques.

Legitimate Traffic Analysis		
No. of Runs	Data Traffic (Mb)	
	Adaptive SFaDMT Technique	Pushback Technique
1st Run	1.0	1.2
2nd Run	1.3	1.8
3rd Run	1.4	1.75
4th Run	1.5	1.9
5th Run	1.7	1.9
6th Run	1.9	1.9
7th Run	1.65	1.75
8th Run	1.9	2.1
9th Run	1.7	1.9

**Table 3 sensors-23-06574-t003:** Detection comparison of both techniques.

Detection Comparison		
No. of Runs	Data Traffic (Mb)	
	Adaptive SFaDMT Technique	Pushback Technique
1st Run	1.0	10.0
2nd Run	1.0	14.0
3rd Run	1.0	15.0
4th Run	3.0	20.0
5th Run	14.0	43.0
6th Run	11.0	63.0

## Data Availability

Not applicable.

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
