# Peer review of "An Adaptive Distributed Denial of Service Attack Prevention Technique in a Distributed Environment"

_sensors, 2023, doi:10.3390/s23146574_

Round 1
Reviewer 1 Report
The novelty of the article is confirmed by the trend of increasing number of DDoS attacks. Effective execution of such attacks leads to large economic losses. The article discusses one type of such attacks that is SYN Flood.
The article shows the relevance of ongoing research. The mechanisms of construction and impact of DDoS attacks are evaluated. A comparative analysis of known methods to detect and counter such attacks was conducted. Their main drawbacks are indicated. To address these shortcomings, the authors propose to use the developed methodology called SYN Flood Attack Detection and Mitigation Technique (SFaDMT). The use of this methodology allows to improve the efficiency of detection and mitigation of this type of attacks. This technique is based on a signature detection method of DDoS attacks. It allows to filter out of the incoming traffic SYN-packets, which include non-existent IP-addresses. If the result matches 70% of the available signatures, this malicious traffic will be mitigated and isolated. In addition, the methodology allows for an adapted solution to be produced. This is achieved through the use of:
- application of a multi-deposit preceptor in a centralized environment;
- pattern and signature-centered strategies that are used to identify DDoS intrusion.
The article presents rules and algorithms to filter out SYN Flood malicious packets.
To evaluate the effectiveness of the developed SFaDMT detection technique, the authors have created a simulation model based on the OMNET++ simulator, which allows creating networks containing from 20 to 300 nodes/clusters. Using this simulator, the authors generated bursts of traffic caused by malicious SYN Flood packets. A comparative analysis was conducted with the Pushback method for distributed denial-of-service attack protection. In the comparative analysis, the authors varied the number of clusters in the network from 8 to 300. The developed SFaDMT method detected and blocked the attack faster than the Pushback method. So for a network consisting of 8 nodes, the Pushback technique detected a DDoS attack of 10 MB. At the same time, SFaDMT detects an attack when only 1 MB of traffic is generated. When the network size is increased to 300 nodes, the Pushback method detects a DDoS attack when 60-63 MB of traffic is generated, while the SFaDMT technique detects an attack when 11-12 MB of traffic is generated. The tests showed that the use of the developed methodology can improve the detection rate of SYN Flood attacks by 26% compared to the Pushback method.
The simulation results show that the objective of the article has been achieved.
Remarks.
1. The article lacks material that allows to estimate the time cost of attack detection. An explanation of how the authors obtained this value is required.
2. As the size of the simulation network increased, the amount of traffic required to detect DDoS attacks using the Pushback method increased steadily. So using 8 nodes requires 10 MB, using 120 nodes requires 40-43 MB, and using 300 nodes requires 60-63 MB of traffic. A similar situation of increasing traffic volume was observed when applying SFaDMT methodology when changing the number of nodes from 8 to 120. However, when simulating network consisting of 300 nodes, the traffic volume decreased to 11-12 MB. The authors need to explain these results.
3. Figure 1 is missing in the article.
4. On page 8, line 297, in the word "...ule 3", the letter R is missing.
5. On page 14, line 446, the authors write "...The detection comparison between the adaptive and SFaDMT technique....". It is necessary to replace the word "adaptive" with the word "Pushback".
Moderate changes in English required
Author Response
First of all thanks for your valuable comments, I tried my level best to incorporate all the highlighted comments in the final version. Submitted for your final approval. Regards

Reviewer 2 Report
This paper investigates the detection issues against DDoS in distributed environments. It focuses on an adaptive attack protection technique. The proposed model is simulated and analyzed in the OMNET simulator and a comparison are made with the latest technique. The authors claim that the proposed method offers improvements in detection performance.
Overall, the topics studied in this manuscript are interesting and can bring new contributions to the information security field. To improve the quality of the manuscript, I give the following suggestions.
1. If possible, the Abstract should end with specific data to support the results of the paper.
2. The authors are recommended to recheck the grammatical issues. For example, the tense of "The proposed model simulates and analyzes in OMNET simulator and a comparison has been made with the latest technique." in the Abstract is confusing.
3. The Introduction is clearly written, but I think it could be better, especially in terms of comprehensiveness. In the last decade of research, the classical information security issues have been extended to the physical layer. Researchers may also focus on simultaneous information and physical security in addition to the focus of this paper. I hope the following two articles can give some help to the authors.
--Mode division-based anomaly detection against integrity and availability attacks in industrial cyber-physical systems. Computers in Industry 137 (2022): 103609.
--Security framework for industrial collaborative robotic cyber-physical systems. Computers in Industry 97 (2018): 132-145.
4. Figures 8 - 11 are not very clear. Authors are suggested to replace the figures with vector graphics.
5. The limitations or shortcomings of the proposed scheme are suggested to be stated in Sec. 5 or Sec. 6.
See comment 2 above.
Author Response

(The authors gave the same response as above.)

Round 2
Reviewer 2 Report
The following comments are given in my first round of review. The authors revised only the fourth item, without considering and responding to other comments. I don't see any peer-to-peer responses in the "Author Response File" either.
---------------------
1. If possible, the Abstract should end with specific data to support the results of the paper.
2. The authors are recommended to recheck the grammatical issues. For example, the tense of "The proposed model simulates and analyzes in OMNET simulator and a comparison has been made with the latest technique." in the Abstract is confusing.
3. The Introduction is clearly written, but I think it could be better, especially in terms of comprehensiveness. In the last decade of research, the classical information security issues have been extended to the physical layer. Researchers may also focus on simultaneous information and physical security in addition to the focus of this paper. I hope the following two articles can give some help to the authors.
--Liu, Bin, et al. "Mode division-based anomaly detection against integrity and availability attacks in industrial cyber-physical systems." Computers in Industry 137 (2022): 103609.
--Khalid, Azfar, et al. "Security framework for industrial collaborative robotic cyber-physical systems." Computers in Industry 97 (2018): 132-145.
4. Figures 8 - 11 are not very clear. Authors are suggested to replace them with vector graphics.
5. The limitations or shortcomings of the proposed scheme are suggested to be stated in Sec. 5 or Sec. 6.
The authors are recommended to recheck the grammatical issues. For example, the tense of "The proposed model simulates and analyzes in OMNET simulator and a comparison has been made with the latest technique." in the Abstract is confusing.
